# Revolutionizing Radiotoxicity Management with Mesenchymal Stem Cells and Their Derivatives: A Focus on Radiation-Induced Cystitis

**DOI:** 10.3390/ijms24109068

**Published:** 2023-05-22

**Authors:** Carole Helissey, Sophie Cavallero, Nathalie Guitard, Hélène Théry, Cyrus Chargari, Sabine François

**Affiliations:** 1Clinical Unit Research, HIA Bégin, 69 Avenu de Paris, 94160 Saint-Mandé, France; 2Department of Radiation Biological Effects, French Armed Forces Biomedical Research Institute, Place Général Valérie André, 91220 Brétigny-sur-Orge, France; socavallero@gmail.com (S.C.); nathalie.guitard@def.gouv.fr (N.G.); helene.thery@intradef.gouv.fr (H.T.); cyrus.chargari@aphp.fr (C.C.); sfm.francois@gmail.com (S.F.); 3Department of Radiation Oncology, Pitié Salpêtrière University Hospital, 47-83 Bd de l’Hôpital, 75013 Paris, France

**Keywords:** radiation-induced injury, radiation cystitis, radiotherapy, mesenchymal stem cells, conditioned medium, extracellular vesicles

## Abstract

Although radiation therapy plays a crucial role in cancer treatment, and techniques have improved continuously, irradiation induces side effects in healthy tissue. Radiation cystitis is a potential complication following the therapeutic irradiation of pelvic cancers and negatively impacts patients’ quality of life (QoL). To date, no effective treatment is available, and this toxicity remains a therapeutic challenge. In recent times, stem cell-based therapy, particularly the use of mesenchymal stem cells (MSC), has gained attention in tissue repair and regeneration due to their easy accessibility and their ability to differentiate into several tissue types, modulate the immune system and secrete substances that help nearby cells grow and heal. In this review, we will summarize the pathophysiological mechanisms of radiation-induced injury to normal tissues, including radiation cystitis (RC). We will then discuss the therapeutic potential and limitations of MSCs and their derivatives, including packaged conditioned media and extracellular vesicles, in the management of radiotoxicity and RC.

## 1. Introduction

In 2020, 1,414,259 new cases of prostate cancer and 375,304 related deaths were reported worldwide [1]. Prostate cancer is the most commonly diagnosed pelvic cancers in men, accounting for 20% of all new cancer cases in European men, followed by colorectal cancers, bladder, and kidney cancers [2].

Radiotherapy is an essential primary or adjuvant therapy for prostate cancer [3]. In addition to its curative role, radiotherapy can also be used to palliate symptoms such as pain or bleeding. However, despite its effectiveness, radiation therapy can cause significant damage to normal tissues surrounding the cancer site, leading to radiotoxicity. The severity of radiotoxicity is directly proportional to the radiation dose but also depends on intrinsic and extrinsic conditions specific to each patient (genetic predisposition, concomitant treatments, adequate preparation during treatment sessions, treatment technique used, etc.). The acute or long-term sides effects can sometimes seriously affect patients’ quality of life [4,5].

One of the major organs affected by radiotoxicity is the bladder. Radiation cystitis, an inflammatory condition of the bladder, can cause pain, frequent urination, hematuria, etc. Severe cases of radiation cystitis can result in fibrosis, ulceration, and necrosis of the bladder wall, which can be life-threatening [6].

Several approaches have been investigated to mitigate radiotoxicity, including the use of instillations, radioprotective agents, and hyperbaric oxygen therapy, with limited success rates and various associated side effects [6]. Such approaches are therefore not widespread in clinical practice.

Mesenchymal stem cells (MSCs) have demonstrated their potential in tissue repair and have thus emerged as a promising therapeutic approach for radiation-induced tissue injury due to their anti-inflammatory, anti-fibrotic, and regenerative properties [7,8,9].

In this review, we will summarize the pathophysiological mechanisms of radiation-induced injury to normal tissues, including radiation cystitis (RC). We will then discuss the therapeutic potential and limitations of MSCs and their derivatives, including packaged conditioned media and extracellular vesicles, in the management of radiotoxicity and RC.

## 2. Molecular and Cellular Mechanisms of Radiation-Induced Normal Tissue Damage

Ionizing radiation induces the production of reactive oxygen species (ROS) and reactive nitrogen species (RNS), responsible for oxidative stress and damage to DNA or mitochondria [10]. Indeed, the production of these free radicals leads to the activation of nicotinamide adenine dinucleotide phosphate oxidase (NADPH oxidase), lipoxygenases (LOXs), nitric oxide synthase (NOS), and cyclooxygenases (COXs) which cause DNA damage directly as well as through epigenetic processes (with methylation, histone acetylation, histone methylation, MiRNA production) altering transcription [10]. The production of ROS and RNS leads to cell death signals by causing mitochondrial DNA to dysfunction, but also by the activation of the c-Jun N-terminal kinase cascade and Bcl-2 protein family members [11].

At the tissue level, radiation-induced damage to healthy tissue is related to persistent inflammation, vascular damage, particularly to endothelial cells, and fibrosis through the accumulation of extracellular matrix leading to loss of organ function [12,13,14,15]. Indeed, there is a close interaction between these three processes. After fractionated irradiation, the accumulation of M2 macrophages promotes fibrosis through the production of TGF-b1 and results in an abnormal wound healing response [6,16,17].

The pathophysiology of radiation cystitis remains unclear. However, Zwaans et al. reported that elevated levels of PAI-1, TIMP-1, TIMP-2, HGF, and VEGF-A were detected in prostate cancer survivors that received a diagnosis of radiation cystitis following radiation therapy. Higher levels of these proteins were also found in patients suffering from hematuria or high symptom scores. In urine, however, inflammatory proteins were detected in patients with severe hematuria and end stage radiation cystitis [18]. This reflects the importance of signaling pathways involved in tissue remodeling, with excessive secretion of extracellular matrix responsible for late radiation cystitis and fibrosis. In the early stages, Kopčalić et al. have reported the importance of inflammation in severe acute radiation cystitis, especially when inflammation is overexpressed [19]. Indeed, a positive correlation exists between high baseline IL6 and TGFb1 levels and severe urinary symptoms. Thus, inflammation and fibrosis in the early stages both contribute to the acute RC process.

Preclinical models of RC have shown firstly loss of the entire urothelium, decreased permeability of the urothelium, with a decrease in superficial urothelial cells, decrease in uroplakins, intra-cytoplasmic relocalization of E-cadherin, loss of the glycosaminoglycan (GAG) layer, oedema and dilatation of vessels, and atypical stromal cells. These lesions appear in the acute phase and are exacerbated by inflammatory processes. Pro-inflammatory cytokines such as COX-2 increase in abundance following activation of the NFkB signaling pathway. Meanwhile, radiation-induced vascular lesions develop, including telangiectasia and friable neovascularization. These lesions are the sites at which fibrosis appears in the late phase, with an accumulation of collagen via an increase in the secretion of TGFb1 and the activation of fibroblasts into myofibroblasts. Inflammation, essentially of the Th2 type, and vascular lesions with hemorrhage persist at this stage. Finally, smooth muscle cells disappear, leading to loss of contractility in the bladder muscle [20] (Figure 1).

## 3. Mesenchymal Stem Cells (MSC)

MSCs are a type of stem cell that has garnered much attention in the field of regenerative medicine. MSCs were originally described by Friedenstein et al. [21], who noted the presence of a rare population of non-hematopoietic cells that had the ability to adhere to plastic and form colonies. These cells were later identified as MSCs, which are now known to be present in many other tissues, including adipose tissue, umbilical cord tissue, and dental pulp, as well as endometrium, peripheral blood, skin, placenta, umbilical cord, synovial fluid, and muscles [22,23,24,25,26,27].

MSCs exhibit a set of characteristics that distinguish them from other cell types. They have the ability to differentiate into a variety of cell types, including bone, cartilage, fat, and muscle. They also have a high self-renewal capacity, meaning that they can divide and differentiate multiple times without losing their potency. Additionally, MSCs have the ability to secrete a variety of growth factors and cytokines that are involved in tissue repair and regeneration. These factors can stimulate the proliferation and differentiation of nearby cells and can also modulate immune responses.

### 3.1. Organization and Isolation of MSCs

The International Society for Cellular Therapy (ISCT) has established criteria to define MSCs, which includes their ability to adhere to plastic, express specific surface markers, and differentiate into osteoblasts, adipocytes, and chondrocytes under appropriate conditions [28,29].

In order to ensure that isolated cells meet the criteria established by the ISCT, it is essential for MSCs to be characterized according to their morphology, surface markers, and differentiation potential:

Adherence to plastic. MSCs have a spindle-shaped morphology and can form colonies when cultured.

Plastic adherence is a method in which MSCs are isolated based on their ability to adhere to plastic surfaces. The tissue sample is first washed and digested with enzymes (such as collagenase) to obtain a single-cell suspension, which is then cultured in a medium containing fetal bovine serum (FBS) or human serum. The cells are plated on a plastic surface and allowed to adhere. Non-adherent cells are removed, and the adherent cells are cultured.

Specific surface antigen (Ag) expression. MSCs express specific surface markers such as CD73, CD90, and CD105, and do not express hematopoietic markers such as CD45, CD34, and CD14.

Fluorescence activated cell sorting (FACS) can be used to isolate MSCs based on their surface markers. Since MSCs typically express CD73, CD90, and CD105, but not CD34 and CD45, by using fluorescently labeled antibodies specific to these markers, pure populations of MSCs can be isolated through FACS.

Multipotent differentiation potential. MSCs can differentiate into osteoblasts, adipocytes, and chondrocytes under appropriate conditions.

### 3.2. Properties

The mechanisms of MSC-based therapy for radiotoxicity management involve several key processes, including their immunomodulatory, anti-fibrotic, and angiogenic properties.

#### 3.2.1. Immunomodulatory Properties

MSCs can modulate the immune response in tissues damaged by radiation, which can help to promote tissue repair and regeneration, by both inhibiting the activation of immune cells that contribute to tissue damage and promoting the activation of immune cells involved in tissue repair [30,31]. They can inhibit the activation of immune cells, such as T cells and macrophages, and reduce the secretion of pro-inflammatory cytokines, such as TNF-α and IL-6. This anti-inflammatory effect can help to reduce tissue damage and promote tissue repair [32,33,34,35]. Moreover, they can stimulate the proliferation and differentiation of endogenous stem cells, and promote the secretion of growth factors and cytokines that support tissue repair.

#### 3.2.2. Anti-Fibrotic Properties

MSCs have been shown to have anti-fibrotic properties, which can help to prevent or reduce fibrosis in tissues damaged by radiation. They can inhibit the differentiation of fibroblasts into myofibroblasts, which are responsible for the production of extracellular matrix proteins that contribute to fibrosis. This anti-fibrotic effect can help to preserve tissue function and prevent long-term tissue damage [36]. MSCs also have the ability to reduce fibrosis by decreasing the number of Th17 cells through indoleamine-2,3-dioxygenase (IDO) expression, and also by decreasing the expression of type I collagen [37]. Additionally, MSCs produce several anti-fibrotic factors including IL-1Ra, FGF-2, HGF, and TSG-6 (tumor necrosis factor stimulated gene-6), and have the potential to induce HGF, KGF (keratinocyte growth factor), as well as BMP-7 (bone morphogenetic protein 7) secretion and decrease TGF-β1 and TNF-α [38,39].

#### 3.2.3. Angiogenic Properties

MSCs can promote angiogenesis, the formation of new blood vessels, in tissues damaged by radiation, thus improving tissue perfusion and oxygenation, which are important for tissue repair and regeneration. MSCs can secrete pro-angiogenic factors, such as vascular endothelial growth factor (VEGF), and promote the migration and proliferation of endothelial cells that contribute to angiogenesis [40,41,42].

### 3.3. Clinical Application of MSCs in Radiotoxicity Management

The use of MSCs has been proposed as a potential treatment for radiotoxicity due to their anti-inflammatory and regenerative proprieties. Radiation exposure can lead to an inflammatory response and an upregulation of pro-inflammatory cytokines, which MSCs can suppress, leading to a reduction in tissue damage and an enhancement of tissue repair. Pre-clinical and clinical studies have been undertaken to evaluate the therapeutic effect of MSCs of different origins in radiotoxicity management of different organ lesions.

Radiodermatitis is the most frequent radiation-induced toxicity, ranging from simple redness to skin fibrosis with an aesthetic and/or functional impact depending on the irradiated area [43,44]. Zheng et al. demonstrated that rat bone marrow MSCs reduce inflammation and fibrosis in injured skin and promote repair of acute radioactive skin injury by decreasing the secretion of TGFb1 and prostaglandin E2 and increasing the secretion of stromal cell derived factor-1 [45].

Another study by François et al. showed that MSCs promoted healing of cutaneous lesions induced by irradiation in a xenogenic transplant model. Compared with non-transplanted irradiated controls, the lesions were found to evolve to a less severe degree of radiation dermatitis after MSC transplant [46]. Other preclinical models of radiodermatitis have confirmed the benefit of MSCs through inhibition of pro-inflammatory molecules and TGFb1 [47]. A medical case examined how cadaveric MSCs could be used to treat a necrotic ulcer in a patient who had received a 50–60 Gy dose of radiation therapy for angioma in the right leg. Two years following the treatment, medical professionals noticed a decrease in the size of the ulcer and an improvement in skin quality, providing evidence of the effectiveness of MSC therapy [48].

Radiation-induced lung injury (RILI) is a severe complication of thoracic irradiation with a functional impact ranging from a simple cough to severe respiratory failure, and a mortality rate close to 15% [49]. Liu et al. investigated the protective effects of decorin (DCN)-modified MSCs on RILI. They found that DCN-modified MSCs attenuated histopathologic injuries by increasing proliferation of epithelial cells, reducing lymphocyte infiltration, and inducing interferon-γ expression, while decreasing apoptosis, inhibiting collagen type III α1 expression in pulmonary tissues, decreasing the proportion of regulatory T-cells (Tregs), and inhibiting fibrosis in the later phase [50].

In two reviews, the benefits of MSCs, from different sources, were also reported in management of radiation-induced digestive, cerebral, cardiac, and hepatic toxicity, primarily through immunomodulation and secretion of anti-fibrotic proteins [51,52]

### 3.4. Clinical Application of MSCs in Radiation Cystitis

Bladder irradiation induces oxidative stress, which causes lesions in the urothelium, inflammation of the vascular lesions, and finally fibrosis resulting in urinary symptoms such as pain, frequency, urgency, incontinence, and hematuria [6,53,54,55,56,57]. 

The effectiveness of MSC-based therapy for radiation cystitis has been demonstrated in several preclinical studies.

Imamura et al. were the first to evaluate the therapeutic potential of MSCs in RC [57]. Their aim was to investigate whether bone marrow MSCs (BM-MSCs) implanted into radiation-damaged urinary bladders could regenerate functional bladder tissue. The study was carried out on female Sprague Dawley rats that were exposed to 2 Gy radiation once a week for five weeks, receiving a total of 10 Gy in the pelvic region. Two weeks after the last radiation treatment, 0.5 × 106 BM-MSCs were implanted with a 29-gauge syringe at four different sites in the anterior and posterior bladder walls. The study found that two weeks after the last irradiation, the smooth muscle layers and nerve fibers of the irradiated urinary bladders were disorganized, and the proportion of smooth muscle layer and nerve fiber areas was significantly decreased. They reported a histoarchitectural reconstruction of radiation-injured urinary bladders, after four weeks of injection of BM-MSCs. The smooth muscle layers and nerve fibers in the cell-implanted urinary bladders were reconstructed, and the proportions of each were significantly higher than those in the cell-free injected controls. BM-MSCs have the ability to differentiate into smooth muscle- and nerve-like cells within urinary bladders that had been subjected to radiation. They also had the capacity to promote the growth or development of the remaining host cells that were not damaged by the radiation, including some blood vessel structures that were formed with the host cells. They demonstrated that the injection of BM-MSCs allows the recovery of bladder function in the early stages of radiotherapy-induced lesions [57].

Qiu et al. investigated the protective effects of adipose-derived MSCs (A-MSCs) against radiation-induced bladder injury in a female rat model [55]. A single dose of 20 Gy was delivered with a linear accelerator and 800 μL serum-free DMED medium containing 1 × 106 A-MSC was evenly injected into the muscular layer of bladder with a 25 G needle. They reported that A-MSCs reduced fibrosis and inflammation within the bladder wall. A-MSCs significantly increased the number of blood vessels in the submucosa, and significantly reduced the collagen/muscle ratio in bladder. A-MSCs were shown to have an immunomodulation function in this model, with decreased levels of TNF-α and TGF-β1.

Brossard et al. assessed confirmed the efficacy of A-MSCs in a rat model of chronic radiation cystitis [56]. Female Sprague Dawley rats were irradiated with 40 Gy of localized radiation to induce CRC. A-MSCs were injected intravenously three times every fortnight, at intervals of two weeks, 4.5, 5 and 5.5 months post-irradiation. A-MSCs were found to reduce CRC with a reduction of vascular lesions. For treated irradiated rats, an average of 1.3 ± 0.4 lesions was observed compared to an average of 4.3 ± 2.4 lesions for the untreated irradiated rats. MSCs have a protective effect on the impermeability of the urothelium, limiting the loss of uroplakin III expression induced by irradiation and maintaining its expression over time in treated irradiated rats compared to untreated irradiated rats. In this model, however, the immunomodulatory properties of A-MSCs were not the only factor, as mast cell levels are also involved in urothelial hyperplasia.

In fact, the immunomodulatory and regenerative mechanisms of MSC-based therapy for radiation cystitis are multi-factorial and include:

Anti-inflammatory properties: MSCs can help to reduce inflammation in the bladder tissue. 

Differentiation potential: MSCs have the ability to differentiate into various cell types, including smooth muscle cells and urothelial cells, which are important components of bladder tissue. This differentiation potential may help to promote tissue regeneration and repair.

Homing and engraftment: MSCs have the ability to migrate to sites of tissue injury and inflammation, which may enhance their therapeutic potential in radiation cystitis. Once at the site of injury, MSCs can engraft and differentiate into tissue-specific cells, further promoting tissue repair and regeneration.

### 3.5. Limits of MCS Use in Clinical Practice

MSCs have shown great promise in preclinical and a few clinical studies for a wide range of therapeutic applications and radiotoxicity management. However, there are also significant limitations to the use of MSCs in clinical practice that need to be addressed before these cells can be widely used as a standard therapy [58,59,60].

One major limitation of MSCs is their heterogeneity, both between individuals, female or male, and within the same individual over time [61,62,63]. This makes it difficult to standardize the preparation and administration of MSCs, and it can also impact the consistency and predictability of their therapeutic effects. Additionally, the source of MSCs (including bone marrow, adipose tissue, or umbilical cord) can also influence their properties, and further research is needed to determine the optimal source for different therapeutic applications [64].

Another limitation of MSCs is the risk of immune rejection [65]. While MSCs are generally considered to have low immunogenicity, they can still elicit an immune response in some patients, particularly if they are derived from a different individual. This can limit their efficacy and increase the risk of adverse reactions. Strategies to overcome this limitation include the use of autologous MSCs (derived from the patient’s own tissues) or MSCs modified to reduce their immunogenicity.

A major challenge in the clinical use of MSCs is their potential for tumorigenicity. While MSCs are generally considered to be non-tumorigenic, there is evidence that they can contribute to the development of tumors in certain contexts [59]. This risk is further increased when MSCs are administered in high doses or in combination with other cell types or growth factors [66,67].

Another limitation of MSCs is their potential for differentiation into unwanted cell types. Since MSCs are generally considered to be multipotent (able to differentiate into several different cell types), differentiation can produce unwanted types such as bone or fat cells, which can limit their therapeutic potential [68,69]. Additionally, the conditions under which MSCs are cultured and expanded can also influence their differentiation potential, thus impacting their safety and efficacy.

Furthermore, the process of expanding and culturing MSCs for clinical use is frequently linked to the acceleration of cellular senescence and the reduction of potency [60,70]. The manufacturing and distribution expenses of MSC products are another significant obstacle to their commercial viability [71].

It was once thought that the therapeutic benefits of MSCs were primarily due to their ability to directly interact with other cells or integrate into tissues and take on different roles. However, it is now widely understood that the MSC secretome is of paramount importance and the main mechanism of action in MSC-based therapies is the secretion of paracrine factors [60,72,73,74,75,76,77,78,79].

## 4. Derived Products of Mesenchymal Stem Cells and Radiation Cystitis

MSCs thus exert their therapeutic potential through their paracrine or autocrine action and products derived from MSCs can be used in a variety of medical applications.

Secretome or conditioned medium from MSCs (MSC-CM) includes the entire array of bioactive molecules secreted by MSCs. It is made up of soluble and vesicular components. The soluble portion is rich in immunomodulatory molecules, cytokines, chemokines, growth factors, and apoptotic bodies, while the vesicular component contains extracellular vesicles (MSC-EVs) that are primarily classified by size [80]. The therapeutic potential of MSC-CM has been demonstrated in several areas such as acute and chronic hindlimb ischemia, myocardial infarction, stroke, neurodegenerative diseases, spinal cord injury, male infertility, acute and chronic wounds, acute liver injury/failure, lung injury, periodontal tissue injury, soft tissue injury, and bone defects [80,81,82,83,84,85,86,87]. Studies have also investigated the use of MSC-CM in the treatment of inflammatory arthritis and multiple sclerosis, as well as hair follicle regeneration and fractionated carbon dioxide wound healing [80,81,82,83,84,85,86,87].

In a pre-clinical model of radiation cystitis, Qiu et al. demonstrated that administration of A-MSC conditioned medium led to enhanced bladder function and preserved histology. This, in combination with the limited cell survival within the bladder, provided indirect evidence that the paracrine pathways of A-MSC play a role in preventing radiation-induced bladder dysfunction. The histological findings suggested that conditioned medium increased the number of blood vessels in the submucosal region and reduced muscular fibrosis. Furthermore, enzyme-linked immunosorbent assay (ELISA) results indicated that the expression of TGF-β1 and TNF-α was reduced after treatment with A-MSC conditioned medium [55].

EVs are particles encased by a bilayer of phospholipids, ranging in size from nano- to micro-scale. MSC-EVs are divided into 3 groups according to their size [88].

Exosomes, which are approximately 40–100 nm in size and abundant in CD63, CD9, CD81 and tumor susceptibility gene 101 (Tsg 101), are created through the internal budding of multi-vesicular bodies (MVBs). They also contain significant amounts of annexins, ALG-2-interacting protein X (Alix), clathrin, heat shock proteins, and have low levels of phosphatidylserine (PS). The activation of the cytoskeleton is necessary for their release.

Microvesicles, which range in size from 80–1000 nm and are abundant in integrins, selectins, and CD40 ligands, are formed by budding from the plasma membrane. Their release is dependent on the activation of the cytoskeleton and calcium influx. The lipid bilayer of microvesicles contains an abundance of cholesterol, sphingomyelin, ceramide, and PS.

Apoptotic bodies, which are approximately 1000–5000 nm in size and abundant in PS, nuclear fractions, and cellular organelles, are produced through cell fragmentation during apoptosis. While the involvement of cytoskeleton-associated molecules in the creation and release of apoptotic bodies has been suggested, the mechanism for relocating fragmented DNA remains unclear.

Mesenchymal stem cells transfer their contents, including proteins, RNA, and lipids, via MSC-EV to recipient cells in various ways such as direct membrane fusion, endocytosis, or phagocytosis, as well as through ligand–receptor interaction [89]. Omics analysis reveals that they contain mRNA and miRNA (miR15, miR126, etc.), proteins involved in adipogenesis (KLF7), in folding (Hsp60), in cell adhesion and integrin binding (CALML5, ENPP1, etc.), in extracellular matrix and protein interaction (Galectin, Fibronectin, MMMP14, COL6A1, etc.), in angiogenesis (VEGF, HGF, CXCR4, etc.), in immunomodulation (IGF-1R, IL-1Ra, …), in regulation of apaptosis and survival as well as chemokines (CCL2, CXCL6, etc.), and various growth factors, etc. [89,90,91,92,93].

In light of these properties, there has been growing interest in the biological functions of MSC-EVs, which have been explored in numerous studies, highlighting their potential as a substitute for MSC cell therapy. The therapeutic potential of MSC-EVs in tissue engineering and regeneration has been demonstrated in several areas, such as lung, cardiological, and neurological disease [50,78,94,95,96,97]. 

The use of MSC-EVs can either inhibit the onset of radiation-induced damage or reduce the length of time it takes for the injury to become apparent [51].

Accarie et al. investigated whether MSC-EVs could be used to reduce radiation-related mucosal barrier damage and animal mortality. To test this hypothesis, the researchers administered a total of 600 μg of MSC-EVs intravenously in three 200-μg injections at 6 h, 24 h, and 48 h after nude mouse whole-body irradiation with 10 Gy. The results of the study showed that treatment with MSC-EVs reduced the instantaneous mortality risk by 85%, thus increasing their survival time. This effect could be attributed to the ability of EVs to reduce mucosal barrier disruption. The researchers also found that MSC-EVs improved the renewal of the small intestinal epithelium by stimulating proliferation and inhibiting apoptosis of epithelial crypt cells. Furthermore, EVs reduced radiation-induced mucosal permeability, as evidenced by the preservation of claudin-3 immunostaining at the tight junctions of the epithelium [98]. MSC-EVs also mitigate the damage to the lungs that arises from exposure to radiation by miR-214-3p transfer [99].

In an in vitro model of RC induced by irradiation of human bladder fibroblasts (HUBF) using the small-animal radiation research platform (SARRP), Helissey et al. found that MSC-EVs and MSC-CM had great therapeutic potential in preventing RC. These results were attributed to down-regulation of fibrosis and inflammatory markers and an increase in the secretion of anti-fibrotic cytokines. Their proangiogenic action induces vessel development from HUVEC cells, thus helping to manage bladder vascular lesions induced by irradiation [100] (Figure 2).

The administration of MSC secretome and microvesicles via intravesical application is a promising approach for the prevention of radiation cystitis. In fact, the MSC secretome and microvesicles contain a variety of growth factors, cytokines, and extracellular vesicles that can modulate the immune system, reduce inflammation, and promote tissue repair. The first step of the procedure is to obtain a sample of MSC secretome from a licensed medical provider. Once obtained, the MSC secretome is administered to the patient via an intravesical catheter once a week for eight weeks. During the treatment period, the patient’s symptoms should be closely monitored to evaluate the efficacy of the treatment. After eight weeks of treatment, the patient’s response to the MSC secretome or microvesicules should be evaluated. If there is no improvement in symptoms, the treatment should be discontinued, and the patient should be reassessed to determine if other treatments may be necessary. As a result, the secretome and microvesicules offer a promising alternative to currently available treatments for radiation cystitis, which are often limited in their efficacy and associated with undesirable side effects. 

This procedure will be assessed in mouse model and clinical studies.

### Limits of Use MSC-VE and MSC-CM in Clinical Practice

Despite the promising potential of MSC-EVs and MSC-CM for therapeutic use, there are still several challenges that need to be addressed before they can be widely used in clinical settings [72,78,83,88]. One of the major challenges is developing reliable, reproducible, and robust methodologies for isolating and purifying the therapeutic agent. Large-scale production is also necessary for clinical utility.

Another challenge is identifying ideal sources of MSCs and their conditioned medium during expansion. The clear classification into different subtypes is still under investigation. This is crucial to ensure that only the most effective sub-populations are used for therapeutic purposes [51,60,83].

In addition, further research is required to determine the most effective dosages and methods of administering MSC-EVs and conditioned media in clinical settings [52].

These challenges need to be addressed before MSCs and their derivatives can be widely used for clinical applications.

## 5. Conclusions

In conclusion, MSCs have shown great promise in the management of radiation toxicity and radiation-induced cystitis. The use of MSCs has been shown to be effective in preventing fibrosis, inflammation, and vascular damage, which are the hallmarks of radiation-induced cystitis. However, the development of MSC-based therapies for clinical use still faces several challenges, such as safety concerns, ethical considerations, scalability, variability, and delivery, which limit the use of MSCs for therapeutic purposes.

The paracrine effects of MSCs have been demonstrated to play a crucial role in the amelioration of radiation-induced damage to various organs, including the lungs and bladder.

The preclinical studies referred to in this article provide compelling evidence that MSC-CM and MSC-EVs can attenuate the damage caused by radiation exposure, reduce inflammation and fibrosis, and promote tissue repair. However, further research is needed to optimize their isolation and purification, establish suitable therapeutic doses and optimal administration routes for clinical use, and better understand the mechanisms of their action. With continued investigation and development, MSC-CM and MSC-EVs have the potential to revolutionize radiotoxicity management and greatly improve the quality of life for patients undergoing radiation therapy.

## Figures and Tables

**Figure 1 ijms-24-09068-f001:**
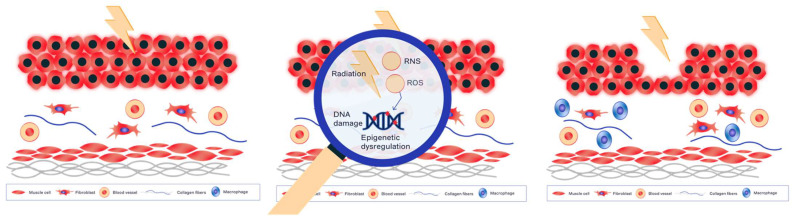
Illustration of early radiation cystitis (RC) and molecular and cellular mechanisms. RNS: reactive nitrogen species, ROS: reactive oxygen species.

**Figure 2 ijms-24-09068-f002:**
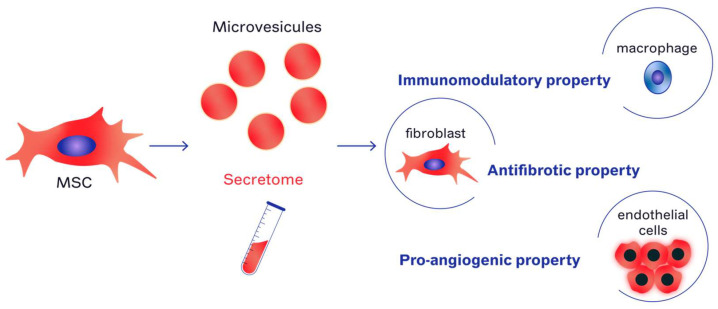
Mechanisms of MSC and their derived (secretome and microvesicules)-based therapy for radiation cystitis.

## Data Availability

Not applicable.

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
