# Peer review of "Revolutionizing Radiotoxicity Management with Mesenchymal Stem Cells and Their Derivatives: A Focus on Radiation-Induced Cystitis"

_ijms, 2023, doi:10.3390/ijms24109068_

Round 1
Reviewer 1 Report
I am wondering why the authors discussed only radiation-induced cystitis instead of whole of body radiation-induced damage. Radiation-induced gut and respiratory system injury is more common and more lethal and so the authors should have discusse dthese issues rather than concentrating on cystitis alone.
How does the MSCs are useful in cystitis-do they secrete any soluble molecules that enhance healing process. Do they produce VEGF, EGF and other factors and if so what are they. Why all patients do not respond to MSCs. What are the causes of failure and causes of success.
What could be the side effects of MSCs.
ok
Author Response
Thanks, reviewers, for your consideration of our article and helpful recommendations.
We have revised the manuscript accordingly.
Best regards,
I am wondering why the authors discussed only radiation-induced cystitis instead of whole of body radiation-induced damage. Radiation-induced gut and respiratory system injury is more common and more lethal and so the authors should have discussed these issues rather than concentrating on cystitis alone.
Answer: We agree with the reviewer that radiation-induced pulmonary and intestinal toxicity are important, and already attract particular attention from clinicians.
Prostate cancer is the most common cancer in men and has an overall survival of more than 80% at 10 years. So, we are seeing a significant proportion of long survivors and quality of life is a key objective. Radiotherapy is an essential primary or adjuvant therapy for prostate cancer is localized external radiation of the prostate. However, radiation-induced bladder toxicity is most often underestimated, although it clearly impacts the quality of life of our patients, especially in prostate cancer patients. Thus, it is essential to limit these toxicities which can lead to radiation cystitis with the risk of cystostomy.
Our article reviewed the pathophysiology of radiation cystitis and the promising therapeutic role of MSC and its derivatives.
How does the MSCs are useful in cystitis-do they secrete any soluble molecules that enhance healing process. Do they produce VEGF, EGF and other factors and if so what are they. Why all patients do not respond to MSCs. What are the causes of failure and causes of success.
What could be the side effects of MSCs.
Answer: MSCs are very good candidates for the management of radiation cystitis due to their immunomodulatory, anti-fibrotic and pro-angiogenic properties. Indeed, in murine models of radiation cystitis, their use allowed regeneration of the urothelium and limitation of vascular damage.
As mentioned, MSCs have pro-angiogenic properties, and vascular regeneration capabilities. As demonstrated in cardiovascular pathologies such as ischemic heart disease, Treatment of Peripheral Artery Disease, and Vascular Bioengineering. Indeed, the positive effects of MSCs on angiogenesis are also attributed to their secretion of HGF, bFGF, IGF-1 and VEGF. In addition, MSCs release exosomes that transfer genetic material and angiogenic molecules, which may promote regenerative processes. These results indicate that the paracrine cytokines and exosomes of MSCs have a complex composition that contributes to their regenerative effects.
The limits of MSCs are as follows:
One major limitation of MSCs is their heterogeneity, both between individuals, female or male, and within the same individual over time (61–63). This makes it difficult to standardize the preparation and administration of MSCs, and it can also impact the consistency and predictability of their therapeutic effects. Additionally, the source of MSCs (including bone marrow, adipose tissue, or umbilical cord) can also influence their properties, and further research is needed to determine the optimal source for different therapeutic applications (64).
Another limitation of MSCs is the risk of immune rejection (65). While MSCs are generally considered to have low immunogenicity, they can still elicit an immune response in some patients, particularly if they are derived from a different individual. This can limit their efficacy and increase the risk of adverse reactions. Strategies to overcome this limitation include the use of autologous MSCs (derived from the patient's own tissues) or MSCs modified to reduce their immunogenicity.
Another limitation of MSCs is the implantation and maintenance of MSCs at the site of the injured tissue.
A major challenge in the clinical use of MSCs is their potential for tumorigenicity. While MSCs are generally considered to be non-tumorigenic, there is evidence that they can contribute to the development of tumors in certain contexts (59). This risk is further increased when MSCs are administered in high doses or in combination with other cell types or growth factors (66,67).
Another limitation of MSCs is their potential for differentiation into unwanted cell types. Since MSCs are generally considered to be multipotent (able to differentiate into several different cell types), differentiation can produce unwanted types such as bone or fat cells, which can limit their therapeutic potential (68,69).
We have included this part in our article.
Reviewer 2 Report
This review, titled "Revolutionizing Radiotoxicity Management with Mesenchymal 2 Stem Cells and Their Derivatives: A Focus on Radiation-Induced Cystitis", presents a novel and interesting perspective.
Throughout the presentation, the authors provided a thorough explanation of the pathophysiology related to radiation. In addition, a detailed explanation of the properties of mesenchymal stem cells as well as their clinical applications was presented.
My recommendation for improving the manuscript would be to add some thoughts regarding the procedures of MSC in the management of radiotoxicity.
In general, the review will be suitable for publication after a minor revision.
There is a need for an overall review and improvement of the English. Reference number 3 should be modified between sentences. It would be more appropriate to replace the "..." with "etc.".
Author Response
Reviewer 2.
Thanks, reviewers, for your consideration of our article and helpful recommendations.
We have revised the manuscript accordingly.
Best regards,
This review, titled "Revolutionizing Radiotoxicity Management with Mesenchymal 2 Stem Cells and Their Derivatives: A Focus on Radiation-Induced Cystitis", presents a novel and interesting perspective.
Throughout the presentation, the authors provided a thorough explanation of the pathophysiology related to radiation. In addition, a detailed explanation of the properties of mesenchymal stem cells as well as their clinical applications was presented.
My recommendation for improving the manuscript would be to add some thoughts regarding the procedures of MSC in the management of radiotoxicity.
In general, the review will be suitable for publication after a minor revision.
There is a need for an overall review and improvement of the English. Reference number 3 should be modified between sentences. It would be more appropriate to replace the "..." with "etc.".
Answer: We thank the reviewer for his comments. We have made the corrections.
The administration of MSC secretome and microvesicles via intravesical application is a promising approach for the prevention of radiation cystitis. In fact, the MSC secretome and microvesicles contain a variety of growth factors, cytokines, and extracellular vesicles that can modulate the immune system, reduce inflammation, and promote tissue repair. The first step of the procedure is to obtain a sample of MSC secretome from a licensed medical provider. Once obtained, the MSC secretome is administered to the patient via an intravesical catheter once a week for eight weeks. During the treatment period, the patient's symptoms should be closely monitored to evaluate the efficacy of the treatment. After eight weeks of treatment, the patient's response to the MSC secretome or microvesicules should be evaluated. If there is no improvement in symptoms, the treatment should be discontinued, and the patient should be reassessed to determine if other treatments may be necessary. As a result, the secretome and microvesicules offer a promising alternative to currently available treatments for radiation cystitis, which are often limited in their efficacy and associated with undesirable side effects.
This procedure will be assessed in mouse model and clinical studies.
We had the article proofread by a native English speaker.